# Drought–Rewatering Cycles: Impact on Non-Structural Carbohydrates and C:N:P Stoichiometry in *Pinus yunnanensis* Seedlings

**DOI:** 10.3390/plants14152448

**Published:** 2025-08-07

**Authors:** Weisong Zhu, Yuanxi Liu, Zhiqi Li, Jialan Chen, Junwen Wu

**Affiliations:** 1The Key Laboratory of Forest Resources Conservation and Utilization in the Southwest Mountains of China Ministry of Education, Southwest Forestry University, Kunming 650224, China; my538076@swfu.edu.cn (W.Z.); lyx1997@swfu.edu.cn (Y.L.); lizhiqi@swfu.edu.cn (Z.L.); chenjialan@swfu.edu.cn (J.C.); 2Key Laboratory of National Forestry and Grassland Administration on Biodiversity Conservation in Southwest China, Southwest Forestry University, Kunming 650224, China

**Keywords:** *Pinus yunnanensis*, repeated drought–rewatering, non-structural carbohydrates, carbon, nitrogen and phosphorus stoichiometric characteristics

## Abstract

The ongoing global climate change has led to an increase in the frequency and complexity of drought events. *Pinus yunnanensis*, a native tree species in southwest China that possesses significant ecological and economic value, exhibits a high sensitivity to drought stress, particularly in its seedlings. This study investigates the response mechanisms of non-structural carbohydrates (NSCs, defined as the sum of soluble sugars and starch) and the stoichiometric characteristics of carbon (C), nitrogen (N), and phosphorus (P) to repeated drought conditions in *Pinus yunnanensis* seedlings. We established three treatment groups in a potting water control experiment involving 2-year-old *Pinus yunnanensis* seedlings: normal water supply (CK), a single drought (D1), and three drought–rewatering cycles (D3). The findings indicated that the frequency of drought occurrences, organ responses, and their interactions significantly influenced the non-structural carbohydrate (NSC) content and its fractions, as well as the C/N/P content and its stoichiometric ratios. Under D3 treatment, stem NSC content increased by 24.97% and 29.08% compared to CK and D1 groups (*p* < 0.05), respectively, while root NSC content increased by 41.35% and 49.46% versus CK and D1 (*p* < 0.05). The pronounced accumulation of soluble sugars and starch in stems and roots under D3 suggests a potential stress memory effect. Additionally, NSC content in the stems increased significantly by 77.88%, while the roots enhanced their resource acquisition by dynamically regulating the C/P ratio, which increased by 23.26% (*p* < 0.05). Needle leaf C content decreased (18.77%) but P uptake increased (8%) to maintain basal metabolism (*p* < 0.05). Seedling growth was N-limited (needle N/P < 14) and the degree of N limitation was exacerbated by repeated droughts. Phenotypic plasticity indices and principal component analysis revealed that needle nitrogen and phosphorus, soluble sugars in needles, stem C/N ratio (0.61), root C/N ratio (0.53), and stem C/P ratio were crucial for drought adaptation. This study elucidates the physiological mechanisms underlying the resilience of *Pinus yunnanensis* seedlings to recurrent droughts, as evidenced by their organ-specific strategies for allocating carbon, nitrogen, and phosphorus, alongside the dynamic regulation of nitrogen storage compounds (NSCs). These findings provide a robust theoretical foundation for implementing drought-resistant afforestation and ecological restoration initiatives targeting *Pinus yunnanensis* in southwestern China.

## 1. Introduction

The frequency of drought events has significantly increased due to dramatic changes in global climate [1]. Concurrently, the spatial and temporal distribution of precipitation has become increasingly uneven, resulting in alternating periods of drought and flooding [2]. Meteorological disasters account for approximately 85% of economic losses caused by natural disasters on a global scale [3]. Drought is recognized as one of the most threatening natural disasters, contributing to roughly 50% of the economic losses attributed to meteorological events [4]. As a widespread abiotic stress, drought has a significant inhibitory effect on plant growth and development [5] and has triggered large-scale tree mortality and forest degradation in several regions worldwide [6]. Conventional studies have primarily examined plant responses to single drought events. However, increasing climate extremes subject plants to recurring droughts interspersed with rehydration periods. These fluctuations in water availability induce variations in plant water balance and carbohydrate/nutrient metabolism, potentially leading to cumulative physiological damage and reduced recovery potential upon rehydration [7]. In the context of widespread drought impacts globally, it is crucial to explore how plants adapt through physiological processes such as regulating carbohydrate metabolism and nutrient element stoichiometry. Research suggests that recurrent droughts lead to more profound metabolic disturbances in plants than single drought events. These disturbances include depletion of carbon reserves [8], sustained impairment of hydraulic function [9], and delayed recovery of growth [10]. Consequently, evaluating the physiological responses of plants to repeated droughts interspersed with rehydration periods holds significant theoretical and practical value. This approach deepens our understanding of the long-term adaptation mechanisms employed by plants and aids in predicting shifts in ecosystem stability.

Non-structural carbohydrates (NSCs), which encompass soluble sugars and starches, play a pivotal role in plant biology by influencing a wide array of biological processes and significantly impacting the growth and development of forest trees [11]. These carbohydrates also serve as a temporary storage form for trees during periods of overproduction [12] and are essential energy sources during growth and metabolic processes [13]. Variations in NSC concentrations frequently indicate the overall carbon availability in plants [14], serve as metrics for evaluating tree growth and survivability [15], and reflect the buffering capacity of trees against external disturbances [16]. Wang et al. [17] demonstrated that the NSC content of *Pinus sylvestris* var. Mongolica seedlings under drought conditions was significantly lower than that of seedlings subjected to normal moisture treatment.

Carbon (C), nitrogen (N), and phosphorus (P) are fundamental elements that constitute the plant body. These elements exert a pivotal influence on various aspects of plant growth, physiological metabolism, and adaptation to environmental changes [18]. Furthermore, they can serve as indicators of the interactions between plants and their environment [19]. The C/N and C/P ratios indicate the efficiency with which plants utilize nitrogen and phosphorus, while the N/P ratio facilitates the evaluation of limiting effects on productivity during plant growth [20]. Research has demonstrated that drought stress elicits varied responses in the C, N, and P ratios of plants [21]. Additionally, the stoichiometric signatures of C, N, and P undergo dynamic changes during drought and rewatering, providing a direct reflection of plant strategies for resource utilization and physiological adaptation [22]. Currently, existing scholarship on the concentration changes and distribution patterns of non-structural carbohydrates (NSCs), along with C, N, and P in trees, primarily focuses on the effects of single drought events. Despite existing research on single drought events, little is known about how repeated drought cycles reshape NSC metabolism and nutrient coordination across plant organs.

*Pinus yunnanensis* is a quintessential native conifer species in southwestern China, widely distributed in the subtropical monsoon climate zone and holding significant ecological and economic value [23]. In recent years, drought has emerged as a critical limiting factor for the growth and development of *Pinus yunnanensis*, particularly during the seedling stage, which is highly sensitive to such stress. Consequently, the mortality rate of seedlings has markedly increased in response to declining rainfall. Yunnan Province, located in the southwestern region of China, has experienced substantial inter-annual precipitation variability and uneven seasonal distribution, resulting in ecological conditions characterized by cyclical drought and alternating wet and dry seasons [24]. A considerable body of research has been conducted on the effects of drought stress on *Pinus yunnanensis*, primarily focusing on changes in growth, morphological characteristics, physiological adaptations, and osmotic regulation under drought conditions [25,26]. In addition, NSCs provide energy for osmotic regulation and C, N, and P reflect nutrient utilization efficiency, which together constitute the key physiological basis for drought adaptation in plants [18]. However, most studies focus on single droughts, lacking insights into how recurrent droughts impact the coordination between NSC metabolism and elemental stoichiometry. The objective of this study was to examine the impact of recurrent drought–rehydration cycles on the dynamics of NSC and C/N/P stoichiometry in *Pinus yunnanensis* seedlings. This investigation was undertaken to address a knowledge gap in research on organ-specific adaptation strategies under repeated stress conditions.

## 2. Results and Analysis

### 2.1. Effects of Repeated Drought–Rewatering on NSC Content and Carbon, Nitrogen, and Phosphorus Stoichiometry of Pinus yunnanensis Seedlings

The effects of organ type, the frequency of drought events, and their interactions on the non-structural carbohydrate (NSC), C, N, and P content, along with their stoichiometric characteristics, in *Pinus yunnanensis* seedlings are presented in Table 1. While the influence of organ type on NSC and C, N, and P concentrations, as well as their stoichiometry, was found to be highly significant (*p* < 0.01), the N content, C/N ratio, and N/P ratio did not show significant changes. The frequency of droughts had highly significant effects (*p* < 0.01) on NSC content and its fractions, C, N, and P concentrations, and their stoichiometry, with the exception of starch, soluble sugars/starch, N content, and N/P ratio; similarly, the interactions between the number of droughts and organ type exhibited highly significant effects (*p* < 0.01) on the aforementioned parameters.

As detailed in Table 1, *Pinus yunnanensis* seedlings demonstrated a dynamic adjustment in the allocation of carbon, nitrogen, and phosphorus resources at the organ level in response to repeated drought–rewatering stress, and the NSC metabolism and elemental stability maintenance are significantly organ-specific.

### 2.2. Effects of Repeated Drought–Rewatering on NSC and Its Components in Pinus yunnanensis Seedlings

As illustrated in Figure 1, the repeated drought–rewatering cycle significantly affected (*p* < 0.05) the dynamics of NSCs in various organs of *Pinus yunnanensis* seedlings. In light of increasing drought occurrences, soluble sugar levels, NSC content, and the soluble sugar/starch ratio initially exhibited an upward trend, followed by a decline. Conversely, the soluble starch content showed a tendency to decrease initially before increasing. However, these fluctuations were not statistically significant (*p* > 0.05). Compared with CK and D1, stem soluble sugar content increased significantly by 32.65% and 30.73%, stem starch content increased significantly by 7.07% and 24.57%, and stem NSC content increased significantly by 24.97% and 29.08% under D3 treatment, and stem soluble sugar/starch ratio did not show any significant difference among treatments (*p* > 0.05). Root soluble sugar content was significantly increased by 60.59% and 72.06%, root NSC content was significantly increased by 41.35% and 49.46%, root soluble sugar/starch ratio was significantly increased by 62.63% and 80.33% (*p* < 0.05), and root starch content was not significantly different among treatments (*p* > 0.05) under D3 treatment compared to CK and D1. Overall, *Pinus yunnanensis* seedlings subjected to repeated drought–rewatering stress demonstrated an active accumulation of NSC in both stems and roots.

### 2.3. Effects of Repeated Drought–Rewatering on C, N, and P Contents and Their Stoichiometric Ratios in Pinus yunnanensis Seedlings

The repeated drought–rewatering treatment significantly impacted the carbon (C), nitrogen (N), and phosphorus (P) contents, as well as their stoichiometric ratios, in each organ of *Pinus yunnanensis* seedlings (Figure 2) (*p* < 0.05). As the frequency of drought events increased, the C content and C/P ratio of needle leaves exhibited a decreasing trend, showing significant reductions compared to CK and D1 treatments under the D3 treatment (18.77% and 16.35%, 24.61% and 20.13%, respectively, *p* < 0.05). Meanwhile, the P content of needle leaves was significantly elevated in the D3 treatment compared to CK and D1 treatments (8.00% and 5.02%, respectively, *p* < 0.05). In contrast, needle leaf N content, C/N ratio, and N/P ratio did not show significant differences among treatments (*p* > 0.05). The content of stem C, as well as the stem C/N and stem C/P ratios, exhibited a gradual increasing trend. In comparison with CK and D1, stem C content demonstrated a significant increase of 77.88% and 50.82% (*p* < 0.05), respectively. The stem C/N ratio exhibited a significant increase of 153.85% and 63.17% (*p* < 0.05), while the stem C/P ratio exhibited a significant increase of 73.58% and 37.99%, under the D3 treatment. However, the stem N and P content, as well as the N/P ratio, did not exhibit a significant difference among the treatments (*p* > 0.05). Root C content, C/N ratio, and C/P ratio initially decreased before increasing, showing significant enhancements under D3 treatment compared to D1 treatment (*p* < 0.05), with increases of 19.72%, 111.68%, and 23.26%, respectively; however, they were not significantly different from CK treatment (*p* > 0.05). The root P content also displayed a decreasing and then increasing trend, showing significant increases under D3 treatment compared to CK treatment (*p* < 0.05) with an 11.68% elevation, but did not significantly differ from D1 treatment (*p* > 0.05). Lastly, root N content and N/P ratio remained statistically unchanged across treatments (*p* > 0.05).

### 2.4. Phenotypic Plasticity Index Analysis of Various Indexes in Repeated Drought–Rewatering Pinus yunnanensis Seedlings

As can be seen in Figure 3, the plasticity indices of 30 indexes, including NSC and its components and the stoichiometric ratios of C, N, and P in *Pinus yunnanensis* seedlings after repeated drought–rewatering alternations ranged from 0.05 to 0.61, with the phenotypic plasticity indices of stem C/N (0.61), root C/N (0.53), root SS/ST, stem C, root SS, and stem C/P being larger; and the phenotypic plasticity indices of root ST, conifer P, conifer C/P, needle P, needle C/P, and stem P had smaller phenotypic plasticity indices. This suggests that *Pinus yunnanensis* seedlings responded to repeated drought–rewatering mainly by changing stem C content and stem C/N, root C/N, and root SS/ST ratios.

### 2.5. Correlation Analysis of Various Indexes of Pinus yunnanensis Seedlings After Repeated Drought–Rewatering

Correlation analysis demonstrated significant relationships between the stoichiometric characteristics of various organs (needles, stems, and roots) and NSC traits (Figure 4). Conifer C content, as well as the C/P and N/P ratios, exhibited significant positive correlations with SS, ST, NSC content, and SS/ST (*p* < 0.05 or *p* < 0.01). Stem C content and the C/P ratio also showed significant positive associations with NSC and its fractions, whereas C/N revealed a weak yet significant positive correlation with SS. Root C content was positively correlated with NSC traits, while root N content and the N/P ratio demonstrated negative correlations with several NSC parameters. In summary, stoichiometric traits associated with C and C/P were found to promote NSC accumulation across different organs, whereas N and N/P ratios exhibited negative correlations primarily in roots. The C and C/P ratios were central to the dominant changes in NSCs, significantly promoting NSC accumulation, particularly in needles and stems.

### 2.6. Principal Component Analysis of Various Indexes of Pinus yunnanensis Seedlings After Repeated Drought–Rewatering

The characteristics of NSCs, along with the C, N, and P content and their respective ratios in the organs of *Pinus yunnanensis* seedlings, were analyzed following a regimen of repeated drought and rewatering using principal component analysis (Figure 5). The cumulative variance contributions from the first two principal components for leaves, stems, and roots were 67.93%, 72.83%, and 71.01%, respectively, indicating that the first two components effectively elucidate the response characteristics of *Pinus yunnanensis* seedlings to repeated drought and rewatering conditions. As illustrated in Figure 5A, the weight coefficients for needle P, needle N content, and the needle N/P ratio were more significant in the first principal component, while the weight coefficients for needle SS, needle NSC, and the needle SS/stem weight ratio were more pronounced in the second principal component. Figure 5B shows that the weight coefficients for stem C, stem SS, and stem C/P were greater in the first principal component, whereas those for stem N/P and stem N were more substantial in the second. In Figure 5C, the weight coefficients for root NSC, root SS, and root C/N were found to be larger, with root ST also exhibiting higher coefficients in the second principal component. In summary, needle P, needle N, needle SS, stem C, stem SS, stem C/P, root NSC, root SS, and root C/N represent the key adaptation characteristics of *Pinus yunnanensis* seedlings to repeated drought and rewatering conditions.

## 3. Discussion

### 3.1. Effects of Repeated Drought–Rewatering on NSC of Pinus yunnanensis Seedlings

The distribution pattern of NSCs in plants is regulated by various physiological and ecological processes. The ratio of starch to soluble sugar content, as well as the overall NSC ratio, exhibit dynamic changes [27]. When plants experience stress, stored NSC serves as a buffer, hydrolyzing to maintain homeostasis. However, if NSC levels fall below a critical threshold, the plant may succumb to carbon starvation [28]. The present study revealed that the frequency of drought events, the specific organs, and their interactions significantly influence NSC content and its constituents in *Pinus yunnanensis* seedlings (Table 1). Under repeated drought–rewatering treatments, the soluble sugar content in various seedling organs demonstrated dynamic alterations. The soluble sugar content was significantly higher than that of CK and D1, with the exception of needle soluble sugar, in D3 (Figure 1A). This suggests that as the number of drought cycles increases, seedlings may gradually establish a more efficient osmoregulatory mechanism [29]. Furthermore, stress memory refers to physiological responses that prepare plants for future stress attacks [30], prioritizing the allocation of soluble sugars to stem and root storage to enhance osmoregulation and expedite recovery following rewatering. The quantity of starch, a principal energy storage compound in plants, is a valuable metric for assessing plant resilience. In this study, stem starch and the NSC contents of stems and roots initially decreased before subsequently increasing with the number of droughts, with D3 exhibiting significantly higher levels than CK and D1 (Figure 1B,C). This pattern suggests that seedlings may initially exhibit lower rates of carbon assimilation, preferentially consuming stored carbon to sustain metabolic functions, resulting in carbon depletion [31]. However, as drought frequency increases, *Pinus yunnanensis* seedlings respond by storing more starch and converting greater amounts of photosynthetic products into starch and NSCs, thereby enhancing resilience to future drought events. Additionally, the root soluble sugar/starch ratio decreased and then increased with the frequency of droughts, with notably higher levels in the D3 treatment compared to CK and D1 (Figure 1D). In contrast, the soluble sugar/starch ratios in needle leaves and stems did not show significant differences among treatments. This could indicate that the root system preferentially accumulates starch for energy reserves under initial drought stress; however, as the number of droughts increases, the demand for soluble sugars to maintain osmotic balance drives accelerated starch hydrolysis into soluble sugars [32]. This finding is consistent with Zhao Nan’s [33] research on *Liquidambar formosana* and *Schima superba* subjected to recurrent drought. All in all, the stable NSC content in needles suggests a conservative strategy, preserving photosynthetic integrity while stems and roots act as storage reservoirs. Moreover, the dynamic NSC allocation to stems and roots under repeated droughts may reflect a form of “stress priming,” where prior drought exposure enhances the seedlings capacity for osmotic adjustment and carbon storage [30]. However, it is important to note that the term “stress memory” here refers to physiological acclimation rather than genetic adaptation, as the experimental duration (50 days) may not be sufficient to induce heritable changes. Future studies could explore this mechanism by measuring gene expression or epigenetic modifications during drought cycles.

In this study, we showed that *Pinus yunnanensis* seedlings exhibit potential stress adaptation in stems and roots during repeated drought–rewatering. The initial drought (D1) led to a substantial decline in stem starch and NSC content, prompting seedlings to adapt by increasing carbon consumption. Following multiple drought events (D3), there was a notable accumulation of soluble sugars, starch, and NSC content in the stems and roots. Additionally, photosynthetic products were preferentially redirected to stems and roots, thereby creating a carbon buffer pool. This accumulation helped prevent seedling mortality due to recurrent droughts. Notably, the NSC fraction in the needles remained stable across treatments, demonstrating a synergistic adaptation in conjunction with the carbon allocation strategies of the stems and roots.

### 3.2. Effects of Repeated Drought–Rewatering on C, N, and P Stoichiometric Characteristics of Pinus yunnanensis Seedlings

The C, N, and P contents in plants reflect their ability to adapt to environmental changes and exhibit heightened sensitivity to external conditions [34]. This study found that the frequency of drought events, various plant organs, and their interactions have different degrees of impact on the C, N, and P contents, as well as their stoichiometric characteristics (Table 1). Carbon serves as a crucial energy source for plant growth and development and constitutes the primary structural component of plant tissues. Nitrogen and phosphorus are essential for synthesizing key compounds, such as photosynthetic pigments and proteins, and play vital roles in nutrient transport and uptake [35]. Notably, in this study, the decrease in needle C content and concurrent rise in root C/P ratio indicate a targeted nutrient shift, possibly facilitating root regeneration under stress. Our observation aligns with Lu Yuan [36], who reported carbon reallocation toward roots in drought-stressed *Robinia*, further substantiating adaptive prioritization. The observed increase in C content may indicate a reduction in seedling growth rate, but it also suggests enhanced resilience against stressors, facilitating better adaptation to drought conditions [37]. Furthermore, the significant rise in stem C content with increasing drought frequency indicates that seedlings may prioritize the allocation of photosynthetically fixed carbon to survival mechanisms over growth in response to stress [38]. This study also revealed that N content in seedling organs did not significantly differ among treatments, whereas root P content was markedly higher than CK under D1 and D3 treatments. This increase may be attributed to drought-induced damage to the root systems of *Pinus yunnanensis* seedlings, necessitating a higher P consumption for repair and leading to enhanced P uptake and utilization from the soil [39]. Additionally, the substantial increase in P content in needles suggests a heightened demand for P to synthesize enzymes and ATP required for photosynthesis following multiple drought stresses, ensuring that seedlings can meet their material and energy needs in anticipation of subsequent droughts [30].

The C/N and C/P ratios reflect the relationship between carbon assimilation and the uptake of nitrogen and phosphorus, thereby revealing the efficiency of nutrient utilization by plants [40]. This study found a high degree of synergy among organ-specific C/N and C/P ratios under repeated drought–rewatering treatments. The stem C/N and C/P ratios increased significantly with the number of droughts, suggesting that stem tissues enhanced drought acclimatization by elevating the proportion of accumulated carbon assimilates. Conversely, the root C/N and C/P ratios initially decreased and then increased, remaining significantly lower than those of the CK and D3 treatments during the D1 treatment. This pattern implies that under drought conditions, seedlings may increase the proportion of P in roots to facilitate growth in adverse environments, thereby improving the absorption of water and nutrients from the soil [41]. And after experiencing multiple drought stresses, seedlings develop an adaptation to drought and exhibit a potential stress memory, allowing this state to be mitigated, although this hypothesis needs to be further verified by molecular or transcriptional analyses. The C/P ratio in conifers was significantly reduced under D3 treatment, likely due to the reallocation of conifer C to the stem and root systems as drought frequency increased. This strategy helps ensure the availability of key physiological resources, such as P, essential for processes like photosynthetic repair and antioxidant functions under drought conditions. The root carbon to phosphorus ratio was significantly higher in group D3 (23.26% increase from D1), which indicated that *Pinus yunnanensis* seedlings preferentially absorbed phosphorus to maintain metabolic activities under repeated drought conditions. This phenomenon was supported by a significant positive correlation between root phosphorus content and NSCs (Figure 4), revealing a trade-off between carbon allocation and nutrient acquisition. Moreover, N and P are critical elements governing the growth and development of land plants, and the N/P ratio can effectively indicate the limiting effects of nitrogen and phosphorus nutrients [42]. The leaf N/P ratio serves as an important marker for assessing nutrient availability for plant growth [20]. According to Klausmeier et al. [43], plant growth is constrained by the N/P ratio: when the ratio is below 14, growth is primarily limited by nitrogen; when exceeding 16, it is constrained by phosphorus; and when ranging between 14 and 16, both N and P limit growth concurrently. In this study, the N/P ratio for each organ progressively decreased with increased drought frequency, without significant differences among treatments. Notably, the N/P ratio for needles remained below 14 across all treatments, indicating that the growth of seedlings under drought conditions is predominantly restricted by nitrogen, and that increased drought frequency may exacerbate nitrogen limitation.

In this study, we demonstrated that recurrent droughts have the potential to intensify nitrogen limitation through two distinct pathways. Firstly, the N/P ratio of needles was consistently less than 14 (Figure 2F), indicating that the nitrogen acquisition capacity did not demonstrate a concomitant improvement with the increase in the number of droughts. Secondly, the C/N ratio of roots exhibited an increase of 111.68% (Figure 2D), suggesting that the allocation of carbon to the root system may have hindered the synthesis of nitrogen uptake-related enzymes [26]. This dual effect may result in a more severe growth limitation of *Pinus yunnanensis* seedlings in natural drought-prone areas.

### 3.3. Main Strategies for Pinus yunnanensis Seedlings to Adapt to Repeated Drought–Rewatering

During their growth, plants continuously adapt to their surrounding environment, resulting in physiological and morphological changes [44]. The phenotypic plasticity of plants serves as a vital mechanism for environmental adaptation and resilience to adverse conditions, reflecting their capacity for growth, development, and adaptability. Additionally, there is a positive correlation between a plant’s ability to adapt to its external environment and the plasticity index [45]. In this study, *Pinus yunnanensis* seedlings showed high plasticity indices for traits like stem C/N (0.61), root C/N (0.53), root SS/ST, etc. These high indices enable *Pinus yunnanensis* seedlings to adapt to repeated droughts. High stem C allows preferential conversion of carbon to soluble sugars (root SS, stem SS, needle SS), which participate in osmotic adjustment to retain water during droughts. Meanwhile, elevated C/N and C/P ratios optimize nitrogen and phosphorus use efficiency, crucial under limited nutrient availability during droughts. This promotes NSC reserves for withstanding drought and recovering during rewatering. Principal component analysis indicates that seedlings mainly adjust needle N and P, needle SS, stem C, stem SS, stem C/N and C/P, root NSC, root SS, and root C/N to adapt to drought–rewatering. In short, the high plasticity of these traits in *Pinus yunnanensis* seedlings is closely related to their drought adaptation mechanisms, guiding future research on plant drought tolerance. Future research could extend this work by (1) investigating the molecular mechanisms underlying stress memory, such as gene expression changes in NSC metabolism; (2) testing the interactive effects of drought frequency and nitrogen addition, given the observed N limitation in this study; and (3) scaling up to field experiments to validate these findings under natural climate variability”.

## 4. Material and Methods

### 4.1. Summary of the Study Area

The experimental site was located in Kunming City, Yunnan Province (E 102°46′, N 25°03′) at an altitude of 1964 m above sea level, characterized by an average annual temperature of 16.5 °C and average annual precipitation of 1035 mm. The region experiences a subtropical plateau monsoon climate, which features a brief frost period and clearly defined wet and dry seasons. Temperatures within the shelter fluctuated between 18.5 °C and 37 °C, with relative humidity levels ranging from 22.3% to 48.0%. The soil used in the experiment was a blend of local red loam and humus, mixed in a volumetric ratio of 3:2. The field capacity (FC) of the soil was determined to be 25.44%, with a bulk density of 1.17 g/cm^3^, total carbon content of 3.26 g/kg, total nitrogen content of 5.98 g/kg, total phosphorus content of 0.62 g/kg, and a pH level of 6.54.

### 4.2. Experimental Materials

The experimental materials consisted of 2-year-old *Pinus yunnanensis* seedlings cultivated at Yiliang Garden Forestry (Good Species No.: Yun R-SS-PY-035-2020). In early March 2024, the seedlings were relocated to the arboretum of Southwest Forestry University, where they were allowed to acclimatize in 60% shade for one week before being transplanted into seedling pots containing 3 kg of homogenized soil. The specifications of the pots included a bottom diameter of 14.5 cm, a top diameter of 20.5 cm, and a height of 18.5 cm. Each pot housed one *Pinus yunnanensis* seedling, and a tray was placed at the bottom. Following transplantation, it was essential to maintain optimal soil moisture to ensure sufficient hydration for the seedlings and facilitate their growth. The experiment was conducted in a light-permeable, well-ventilated awning equipped with mulch to prevent the influence of rainfall and groundwater vapor on the soil moisture levels within the pots. After a three-month dormancy period, 60 *Pinus yunnanensis* seedlings displaying healthy, uniform characteristics were selected in June for the experimental study.

### 4.3. Experimental Design

Based on the findings regarding current climate periodic changes [24] and previous studies on single drought events [41,46], to simulate the increasing frequency of drought events under climate change, we designed three drought–rewatering cycles (D3) as the highest stress treatment. This choice was supported by two lines of evidence: (1) pretests on conifer species showed that 35% FC induced moderate water stress (visible wilting after 7 days), which is ecologically relevant to the seasonal droughts in southwestern China [1]; (2) previous studies on conifers have shown that 2–3 drought cycles can elicit physiological acclimation without causing irreversible damage [9,10]. This experiment was carried out in July 2024. On the basis of a single drought, the repeated drought–rewatering cycle was simulated by potting water control method, and the field capacity (FC) was determined by weighing method.

The experimental design included three distinct treatment groups: a control group (CK), a single drought treatment group (D1), and a three drought–rewatering cycles treatment group (D3) [47]. For each treatment group, 20 biological replications were established, with each replication comprising a single, healthy *Pinus yunnanensis* seedling, resulting in a total of 60 plants. The specific treatment protocols were as follows: the control group (CK) received a normal water supply throughout the experimental period, maintaining soil moisture at 80% ± 5% of FC (actual moisture content 19.08% to 21.62%); the single drought treatment group (D1) maintained soil moisture at 80% ± 5% FC for the first 38 days, after which watering was ceased to allow soil moisture to decline naturally to 35% ± 5% FC (actual moisture content 7.63% to 10.17%) over 4–5 days, followed by 7 days of drought stress. The 35% FC threshold was chosen based on two criteria: (1) it represents the lower limit for *Pinus yunnanensis* seedling survival under short-term drought [46], and (2) it matches the soil moisture during severe drought periods in Yunnan Province [48]. The 7-day duration was determined by pretests showing that this period elicits significant reductions in photosynthesis and NSC accumulation without causing permanent hydraulic damage [26], which is consistent with the 5–10 day drought periods used in similar conifer studies [48]. Seedlings in the three drought–rewatering cycles treatment group (D3) underwent three drought events coupled with two rehydration sessions. In each drought event, soil moisture was permitted to decrease naturally to 35% ± 5% FC, maintained at this level for 7 days, after which the soil was rewatered to 80% ± 5% FC and maintained for an additional 7 days to facilitate seedling recovery [47]. The overall experimental duration was 50 days, with each drought phase lasting 7 days followed by 7 days of rewatering. On the morning following the conclusion of the D3 treatment, five seedlings were sampled from each treatment group, totaling 15 plants. This experiment was based on a 50-day pot trial that emphasized short-term responses, but did not assess long-term genetic adaptation.

### 4.4. Measurement of Indicators

The samples were systematically divided into needles, stems, and roots, then dried at 105 °C for 30 min (enzyme deactivation) and 80 °C to constant weight. Organ biomass was recorded. Pulverized samples were sieved (0.5 mm mesh) for NSC, C, N, and P analysis. NSC quantification (sum of soluble sugars (SS) and starch (ST)):

SS extraction: 0.05 g dry sample + 10 mL 80% ethanol; vortexed, centrifuged (4000 rpm, 10 min). SS assay: supernatant reacted with 0.2% anthrone (in 72% H_2_SO_4_) at 100 °C (10 min). Absorbance measured at 625 nm using a UV-Vis spectrophotometer (Shimadzu UV-1800). Calibration: glucose standards (0–100 μg/mL; R^2^ > 0.99). ST assay: pellet hydrolyzed with 30% perchloric acid (100 °C, 30 min), neutralized (NaOH), and quantified identically to SS [26].

Total Carbon (C): dried samples (0.2 g) combusted at 950 °C in an Elementar vario MACRO cube CN analyzer (Langenselbold, Germany) via dry oxidation. Calibration: acetanilide standards (71.09% C; accuracy ±0.3%). Total nitrogen (N): Micro-Kjeldahl digestion (Kjeltec 8400, FOSS, Hillerød, Denmark): 0.1 g sample + 5 mL conc. H_2_SO_4_ + catalyst (K_2_SO_4_/CuSO_4_, 10:1) at 420 °C for 45 min. Distilled NH_3_ quantified by titration (0.01M HCl). Total phosphorus (P): samples (0.1 g) digested in 10 mL HNO_3_-HClO_4_ (4:1) at 280 °C until clear. P quantified via molybdenum-blue method [49]: Reagent 0.5% ammonium molybdate + 0.1% antimony potassium tartrate in 2.5M H_2_SO_4_ + 10% ascorbic acid; detection: Shimadzu UV-1800 at 700 nm; calibration: KH_2_PO_4_ standards (0–5 ppm; *R^2^ = 0.998*)

Quality control: analytical blanks and certified reference materials (NIST 1547 Peach Leaves: 1.57% N, 0.17% P) were included in each batch. Recovery rates: 95–103% for N, 92–98% for P.

### 4.5. Data Analysis and Statistics

The normality and homogeneity of the data were assessed using the Kolmogorov–Smirnov test prior to the implementation of subsequent statistical analysis. Subsequently, multiple comparisons were made between the means of the data using the Duncan test. The present study investigated the correlation between light wood growth indices and leaf physiological indices using Pearson correlation analysis. All statistical analyses were performed using SPSS 20.0 (IBM SPSS Statistics, Armonk, NY, U.S.A.), with the level of statistical significance set at *p* = 0.05. The generation of plots was facilitated by utilizing Origin 2021 software.

Plasticity index: P = (X_max_ − X_min_)/X_max_, where X_max_ and X_min_ denote the maximum and minimum values of each indicator [46].

## 5. Conclusions

This study investigated the non-structural carbohydrates and the stoichiometric characteristics of carbon, nitrogen, and phosphorus in *Pinus yunnanensis* seedlings by simulating repeated drought and rehydration cycles in a controlled potting experiment. The results indicated that the number of drought events, plant organs, and their interactions significantly impacted NSC content and its components, as well as the carbon, nitrogen, and phosphorus content and stoichiometric characteristics of the seedlings. Notably, stems and roots exhibited stress memory effects during repeated drought–rewatering cycles. In the D3 treatment, NSCs in stems and roots accumulated significantly, with soluble sugars, starch, and NSC contents increasing by 32.65% to 80.33% compared to CK. This suggests that the stems and roots enhance osmotic adjustment and drought recovery potential through the active storage of carbon resources. Furthermore, stem carbon content and the C/P ratio were significantly elevated (by 77.88% and 73.58%, respectively), while needle and leaf carbon content decreased by 18.77%. Conversely, P absorption increased (8%), and the C/N and C/P ratios of each organ exhibited synergistic variations, reflecting adaptive adjustments to repeated droughts. Seedling growth was primarily limited by nitrogen (needle N/P < 14), and the level of nitrogen limitation intensified with an increasing number of droughts. Combined with the phenotypic plasticity index and principal component analysis, the findings demonstrated that *Pinus yunnanensis* seedlings primarily adapted to repeated drought–rewatering cycles by adjusting the indices of needle nitrogen and phosphorus, needle soluble sugars, stem carbon, stem soluble sugars, stem C/N and C/P ratios, as well as root NSC, root soluble sugars, and root C/N ratios. This research provides insights into the adaptation strategies of *Pinus yunnanensis* seedlings with respect to NSC and carbon, nitrogen, and phosphorus stoichiometric characteristics during repeated drought and recovery, thus offering a theoretical basis for future *Pinus yunnanensis* silviculture and management. Future investigations could include increasing the number of drought–rewatering cycles or appropriately adding nitrogen to observe the adaptive mechanisms of plants in extreme environments over the long term.

## Figures and Tables

**Figure 1 plants-14-02448-f001:**
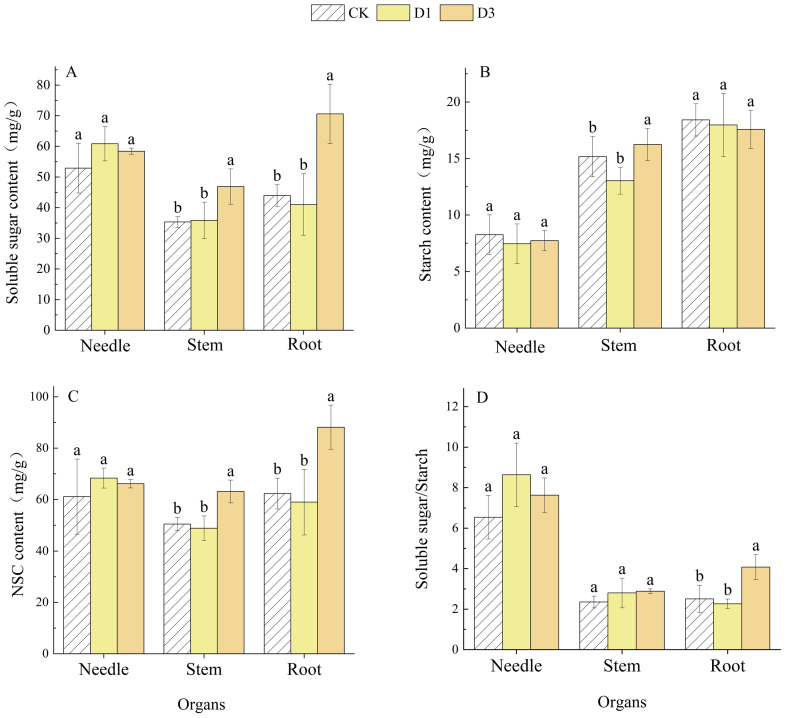
NSC content of each organ of *Pinus yunnanensis* seedlings under repeated drought–rewatering. Note: (**A**) soluble sugar; (**B**) starch; (**C**) NSC; (**D**) soluble sugar/starch; CK: control; D1: single drought-treated group; D3: three drought-treated group; different lowercase letters on the columns indicate that there are significant differences between treatments for the same organ, *p* < 0.05. Same below.

**Figure 2 plants-14-02448-f002:**
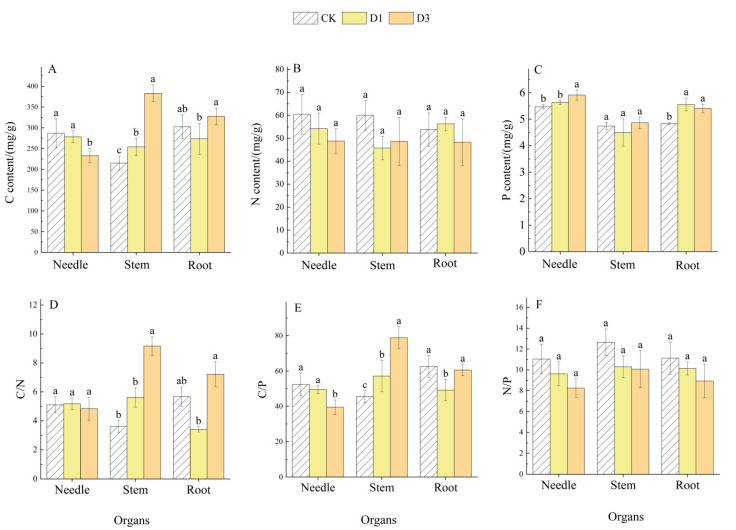
C, N, and P contents and their ratios in various organs of *Pinus yunnanensis* seedlings under repeated drought–rewatering. Note: (**A**) C content; (**B**) N content; (**C**) P content; (**D**): C/N; (**E**): C/P; (**F**) N/P;CK: control; D1: single drought-treated group; D3: three drought-treated group; different lowercase letters on the columns indicate that there are significant differences between treatments for the same organ, *p* < 0.05. Same below.

**Figure 3 plants-14-02448-f003:**
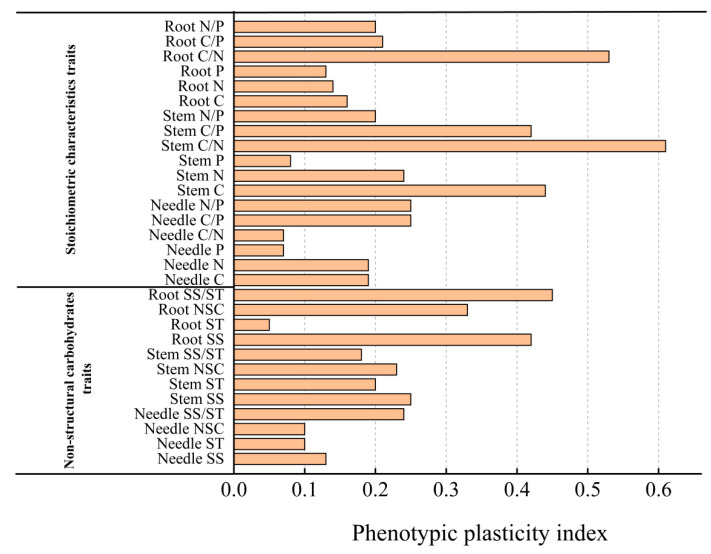
NSC characteristics of different organs and analysis of phenotypic plasticity indices of C, N, P, and their ratios in *Pinus yunnanensis* seedlings after repeated drought–rewatering.

**Figure 4 plants-14-02448-f004:**
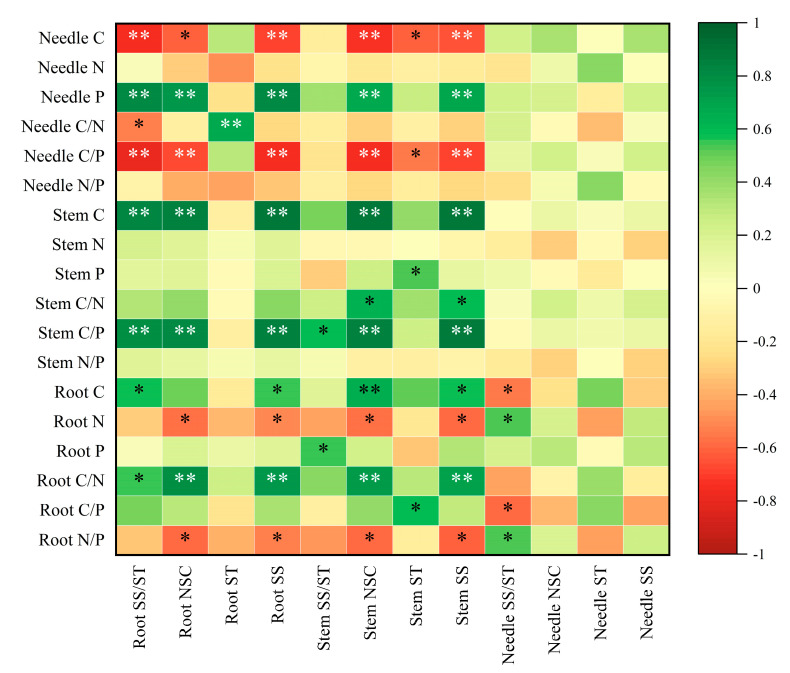
Correlation analysis of NSC with C, N, and P contents and their ratios in each organ of *Pinus yunnanensis* seedlings under repeated drought-restoration water. Note: Green represents a positive correlation; the darker the green, the stronger the positive correlation (values close to 1); the lighter the color, the weaker the positive correlation. White indicates a correlation approaching 0, meaning there is no significant relationship between the variables. Brown represents a negative correlation; the darker the brown, the stronger the negative correlation (values close to −1); the lighter the color, the weaker the negative correlation. * indicates *p* < 0.05, ** indicates *p* < 0.01.

**Figure 5 plants-14-02448-f005:**
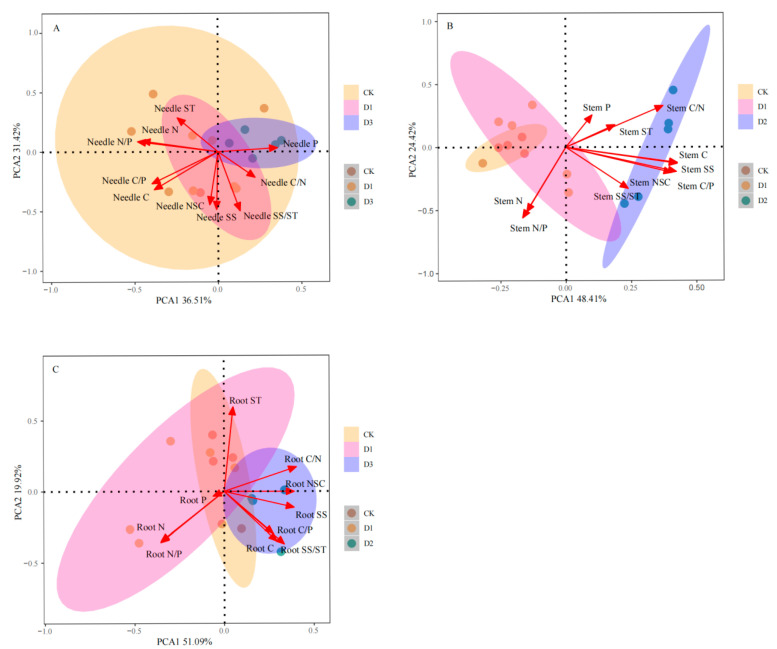
Principal component analysis of NSC characteristics of different organs and C, N, and P contents and their ratios in *Pinus yunnanensis* seedlings after repeated drought–rewatering. Note: The color of the dots in the figure represents the different treatments; the ellipses are the confidence intervals of each parameter for the different treatments; and the arrows represent the relationship between each indicator and the principal components. (**A**) Needle; (**B**) stem; (**C**) root.

**Table 1 plants-14-02448-t001:** Two-way ANOVA of the effect of drought number and organ on NSC content and carbon, nitrogen, and phosphorus stoichiometry of *Pinus yunnanensis* seedlings.

Indicators	Organs		Number of Droughts		Organs × Number of Droughts	
	F	P	F	P	F	P
Soluble sugar	24.076 **	0.000	17.724 **	0.000	6.388 **	0.001
Starch	86.752 **	0.000	1.259	0.296	0.978	0.432
NSC	16.411 **	0.000	16.827 **	0.000	5.467 **	0.002
Soluble sugar/starch	76.106 **	0.000	2.991	0.063	2.077	0.104
C	7.762 **	0.002	17.659 **	0.000	29.811 **	0.000
N	2.323	0.113	2.376	0.107	3.072 *	0.028
P	68.042 **	0.000	10.222 **	0.000	5.773 **	0.001
C/N	1.620	0.212	9.339 **	0.001	5.538 **	0.001
C/P	23.650 **	0.000	8.051 **	0.001	26.874 **	0.000
N/P	2.159	0.130	2.523	0.094	2.509	0.059

Note: F indicates the effect of repeated drought treatments on each indicator, * indicates *p* < 0.05, ** indicates *p* < 0.01. Same below.

## Data Availability

The datasets used during the current study are available from the corresponding author upon reasonable request.

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
