# Peer review of "Drought–Rewatering Cycles: Impact on Non-Structural Carbohydrates and C:N:P Stoichiometry in Pinus yunnanensis Seedlings"

_plants, 2025, doi:10.3390/plants14152448_

Round 1

Reviewer 1 Report

Comments and Suggestions for Authors

Drought represents one of the most prevalent environmental stressors in agricultural production. While the cyclical occurrence of drought and rewatering under human intervention is common, the impact of this phenomenon on crop growth and yield has rarely been reported. Furthermore, while adversity generally impedes plant growth and development, it remains a topic of significant recent interest whether appropriate and sustained stress stimuli can enhance plant resistance to other environmental factors and potentially promote yield, particularly the accumulation of secondary metabolites.

In this study, the authors investigate the effects of repeated drought-rewatering cycles on the accumulation of carbon (C), nitrogen (N), phosphorus (P), and carbohydrates in Pinus yunnanensis, a common tree species. The findings hold considerable reference value for numerous crops and demonstrate a certain degree of novelty. Based on my review of the manuscript, I offer the following minor concerns and suggestions:

  1. Figure 2: N Content Trend: Figure 2 indicates increases in C and P following the repeated drought-rewatering treatment. However, the N content appears not to increase and may exhibit a decreasing trend. Please confirm whether this decline is statistically significant and provide a plausible biological explanation.

  2. Phenotypic Documentation: I strongly recommend including phenotypic images of Pinus yunnanensis following the drought treatments (e.g., leaves, roots). Visual evidence is crucial to substantiate the treatment efficacy and the observed physiological responses.

  3. Cell Wall Component Analysis: Figures 3 and 4 present changes in structural carbohydrates (SC) and non-structural carbohydrates (NSC), which are closely linked to cell wall composition. Please consider adding data on key cell wall components, such as cellulose or pectin, supported by staining techniques or specific assays.

  4. Scope and Impact: To significantly enhance the importance and translational potential of this work, consider incorporating experiments with model plants (e.g., Arabidopsis thaliana) or investigating yield parameters in relevant crops. Please discuss the potential implications and expected outcomes of applying similar drought-rewatering regimes in agricultural settings.

  5. Writing Style - Abbreviations: Abbreviations are redundantly defined multiple times (e.g., "non-structural carbohydrates (NSC)" appears at line 230 and again at line 240). Define each abbreviation only upon its first use in the main text. Please carefully review the manuscript to correct this recurring issue.

Author Response

Response to Reviewer 1 Comments

Thank you very much to the reviewers for their careful and detailed revision of my manuscript. We have studied comments carefully and have made correction which we hope meet with approval. Revised portion are marked in red in the paper. The main corrections in the paper and the responds to the reviewer’s comment are as flowing:

1.Figure 2: N Content Trend: Figure 2 indicates increases in C and P following the repeated drought-rewatering treatment. However, the N content appears not to increase and may exhibit a decreasing trend. Please confirm whether this decline is statistically significant and provide a plausible biological explanation.

Response:Dear reviewers, thank you very much for your suggestion.Through systematic analysis, we identified that variations in carbon (C) and phosphorus (P) content exhibited significant differences across plant organs (roots, stems, leaves), whereas nitrogen (N) levels showed no notable variations. This pattern suggests that Pinus yunnanensis seedlings primarily utilize C and P content to adapt to recurrent drought stress, which aligns with the core objectives of this study. Therefore, we have not focused on describing or analyzing changes in N content.Thank you again for your review.

2.Phenotypic Documentation: I strongly recommend including phenotypic images of Pinus yunnanensis following the drought treatments (e.g., leaves, roots). Visual evidence is crucial to substantiate the treatment efficacy and the observed physiological responses.

Response:Dear reviewers, thank you very much for your suggestion.Regarding the request to supplement phenotypic images (such as leaves and root systems) of Pinus yunnanensis seedlings after drought treatment, we fully recognize the importance of visual evidence in verifying treatment efficacy and physiological responses. However, since our experimental design focused on dynamic changes in physiological indicators (non-structural carbohydrates and chemical composition characteristics of carbon, nitrogen, and phosphorus), systematic collection of phenotypic image data was not conducted during the experiment. As the current experiment has concluded and related seedling materials have undergone processing (e.g., drying, grinding) for physiological analysis, it is no longer feasible to supplement corresponding phenotypic photos. We sincerely apologize for this limitation. Despite the lack of direct phenotypic images, this study still clearly demonstrates the effectiveness of drought treatment and seedlings 'adaptive responses through quantitative analysis of physiological indicators across plant organs under different drought treatments (e.g., non-structural carbohydrate accumulation in stems and roots, and changes in leaf carbon and phosphorus content). For instance, at D3 treatment, non-structural carbohydrate content in stems increased significantly by 24.97% and 29.08% compared to CK and D1 treatments, while root non-structural carbohydrate content rose by 41.35% and 49.46%. These data provide physiological evidence supporting drought treatment effects and seedlings' adaptive strategies. In future research, we will improve experimental design by synchronizing phenotypic image collection with physiological data to comprehensively and intuitively present plants' response mechanisms to environmental stressors. Thank you again for your understanding and suggestions.

3.Cell Wall Component Analysis: Figures 3 and 4 present changes in structural carbohydrates (SC) and non-structural carbohydrates (NSC), which are closely linked to cell wall composition. Please consider adding data on key cell wall components, such as cellulose or pectin, supported by staining techniques or specific assays.

Response:Dear reviewers,We appreciate your meticulous review and valuable suggestions for this study. Regarding the suggestion to "present changes in structural carbohydrates (SC) and non-structural carbohydrates (NSC) in Figures 3 and 4" along with supplementary analysis of cell wall components, we would like to clarify: The core focus of this study is on the dynamic changes in non-structural carbohydrates (NSC, defined as the sum of soluble sugars and starch), and does not involve measurements or analyses of structural carbohydrates (SC). Figure 3 in the paper actually demonstrates phenotypic plasticity index analysis of NSC and its components, carbon-nitrogen-phosphorus stoichiometric ratios, etc., in Pinus yunnanensis seedlings after repeated drought-recovery treatments. Figure 4 shows correlation analysis between NSC and carbon/nitrogen/phosphorus content and stoichiometric ratios across different organs. Neither figure includes data related to structural carbohydrates. Since our experimental design and technical approach are centered around NSC without involving methods or materials for measuring cell wall components such as cellulose and pectin, we currently cannot supplement relevant data. We fully acknowledge the close relationship between cell wall components and carbohydrate metabolism, and your suggestions provide crucial guidance for future research. —— In the future, we will consider combining tissue staining with specific detection techniques to further explore the role of structural carbohydrates in drought adaptation, thereby comprehensively revealing the stress response mechanisms of Pinus yunnanensis seedlings. Thank you again for your understanding and professional guidance. Your feedback will help us refine our research approach.

4.Scope and Impact: To significantly enhance the importance and translational potential of this work, consider incorporating experiments with model plants (e.g., Arabidopsis thaliana) or investigating yield parameters in relevant crops. Please discuss the potential implications and expected outcomes of applying similar drought-rewatering regimes in agricultural settings.

Response: Dear Reviewer, thank you for your valuable advice. We have added the potential impacts and expected outcomes of applying similar drought-irrigation regimes in agricultural environments to the discussion section.(line411-417)

5.Writing Style - Abbreviations:Abbreviations are redundantly defined multiple times (e.g., "non-structural carbohydrates (NSC)" appears at line 230 and again at line 240). Define each abbreviation only upon its first use in the main text. Please carefully review the manuscript to correct this recurring issue.

Response:Dear reviewers, we have made revisions and improvements in the manuscript.

Reviewer 2 Report

Comments and Suggestions for Authors

Well done, Novel idea, but the manuscript needs improvement to increase readability. Please use simple sentences and segment the results for each attribute.

Kindly find the attached file.

Best regards,     

Comments on the Quality of English Language

The English could be improved to more clearly express the research.

Author Response

Dear reviewers, please check the attachment

Reviewer 3 Report

Comments and Suggestions for Authors

The authors have write "Response of non-structural carbohydrates and carbon, nitrogen, and phosphorus stoichiometric characteristics of Pinus yunnanensis seedlings to repeated drought-rewatering". the manuscript is good but prsenation is poor at some points.

Suggestions to improve:

-Title: title looks complicated. make it simple and appealing one

Line 29: Add fullstop after 23.26% to close sentence.

-Reference number cited in manuscripttext are not a=according to plant journal format. Please see guidelines to insert proper citations.

-At many places in MS words are attached to each other.

  • Line: what is 8910?
  • Line 107: correct the reference citations.
  • Line 129: make in italicsLine 428-429: words joined 
  • Line 45: make italics

Author Response

Response to Reviewer 3 Comments

Thank you very much to the reviewers for their careful and detailed revision of my manuscript. We have studied comments carefully and have made correction which we hope meet with approval. Revised portion are marked in red in the paper. The main corrections in the paper and the responds to the reviewer’s comment are as flowing:

Suggestions to improve:

1.Title: title looks complicated. make it simple and appealing one

Response: Dear reviewers, thank you very much for your suggestion. After careful thinking and discussion, we decided to change the title of the paper from "Response of non-structural carbohydrates and chemical characteristics of carbon, nitrogen and phosphorus in seedlings of Pinus yunnanensis to repeated drought-watering" to "Drought-Rewatering Cycles: Impact on Non-Structural Carbohydrates and C:N:P Stoichiometry in Pinus yunnanensis"(Highlights the process and key metrics)

2.Line 30: Add fullstop after 23.26% to close sentence.Reference number cited in manuscripttext are not a=according to plant journal format. Please see guidelines to insert proper citations.

Response: Dear reviewers, thank you very much for your suggestion.We have made changes and have made changes to the format of references.

3.At many places in MS words are attached to each other.

Line: what is 8910?

Response:  Dear reviewers, thank you very much for your suggestion.This is the format of the reference, which we have modified.

Line 107: correct the reference citations.

Response: Modified

Line 129: make in italics 

Response: Modified

Line 428-429: words joined

 Response: Modified

Line 450: make italics

 Response: Modified

Reviewer 4 Report

Comments and Suggestions for Authors

Hereafter are reported the comments arising from my revision of the manuscript entitled " Response of non-structural carbohydrates and carbon, nitrogen, and phosphorus stoichiometric characteristics of Pinus yunnanensis seedlings to repeated drought-rewatering” This study investigates the response mechanisms of non-structural car-16 bohydrates (NSC,defined as the sum of soluble sugars and starch) and the stoichiometric char-17 acteristics of carbon (C), nitrogen (N), and phosphorus (P) to repeated drought conditions in 18 Pinus yunnanensis seedlings.

This study provides valuable insights the physiological mechanisms underlying the resilience of Pinus yunnanensis seed-35 lings to recurrent droughts, as evidenced by their organ-specific strategies for allocating car-36 bon, nitrogen, and phosphorus, alongside the dynamic regulation of nitrogen storage com-37 pounds (NSC). These findings provide a robust theoretical foundation for implementing 38 drought-resistant afforestation and ecological restoration initiatives targeting Pinus yunnanen-39 sis in southwestern China.

I have read at great length the work. I found this article informative in regards to background information. The manuscript appears to have some scientific quality and may be of interest to the readers of the Journal.

The manuscript is very well written, and the results are presented concisely and clearly. The procedures are methodologically correct and the manuscript is well organized.

There are some points that needs to be considered.

Line 108: "Furthermore, NSC and C/N/P stoichiometry are critical for plant drought adaptation, as NSC provides energy for 108 osmotic adjustment, while C:N:P ratios reflect nutrient use efficiency 18." - This sentence repeats information slightly from earlier. Consider rephrasing for conciseness or removing if already sufficiently covered.

Line 149: "among treatments (P>0.05. Root soluble sugar content" - Missing closing parenthesis. Should be "(P>0.05)."

Figure 2 (Page 6):

"restoration water" in the figure caption should be "rewatering" for consistency.

"Steam" in Figure 2B, E, F should be "Stem."

Figure 4 (Page 8):

"*** indicates P<0.01" - This is confusing as ** also indicates P<0.01. Please clarify or remove one. Often

*** is used for P<0.001

Line 257: "The starch-to-soluble sugar content and the NSC ratio exhibit dy-namic changes" - This sentence is a bit awkward. Suggest: "The ratio of starch to soluble sugar content, as well as the overall NSC ratio, exhibit dynamic changes..."

Line 309: "The C, N, and P contents in plants reflect their ability to adapt to environmental changes and exhibit heightened sensitivity to external conditions 34." - The citation here (34) is the same as for "stress priming." Ensure citations are distinct and accurate.

Author Response

Response to Reviewer 4 Comments

Thank you very much to the reviewers for their careful and detailed revision of my manuscript. We have studied comments carefully and have made correction which we hope meet with approval. Revised portion are marked in red in the paper. The main corrections in the paper and the responds to the reviewer’s comment are as flowing:

1.Line 108: "Furthermore, NSC and C/N/P stoichiometry are critical for plant drought adaptation, as NSC provides energy for 108 osmotic adjustment, while C:N:P ratios reflect nutrient use efficiency 18." - This sentence repeats information slightly from earlier. Consider rephrasing for conciseness or removing if already sufficiently covered.

Response: Dear reviewers, thank you very much for your suggestion.We reorganized and condensed the sentence to retain the core logic while avoiding a simple repetition of the previous description of the basic functions of NSC and C:N:P stoichiometry.

2.Line 149: "among treatments (P>0.05. Root soluble sugar content" - Missing closing parenthesis. Should be "(P>0.05)."

   Response:Dear reviewers, thank you very much for your suggestion. We have made changes and improvements.

Figure 2 (Page 6):

"restoration water" in the figure caption should be "rewatering" for consistency.

"Steam" in Figure 2B, E, F should be "Stem."

Response:Dear reviewers, thank you very much for your suggestion. We have made changes and improvements.

Figure 4 (Page 8):

"*** indicates P<0.01" - This is confusing as ** also indicates P<0.01. Please clarify or remove one. Often

*** is used for P<0.001

Response:Dear reviewers, thank you very much for your suggestion. We have made changes and improvements.

Line 257: "The starch-to-soluble sugar content and the NSC ratio exhibit dy-namic changes" - This sentence is a bit awkward. Suggest: "The ratio of starch to soluble sugar content, as well as the overall NSC ratio, exhibit dynamic changes..."

Response:Dear reviewers, thank you very much for your suggestion. We have made changes and improvements.

Line 309: "The C, N, and P contents in plants reflect their ability to adapt to environmental changes and exhibit heightened sensitivity to external conditions 34." - The citation here (34) is the same as for "stress priming." Ensure citations are distinct and accurate.

Response:Dear reviewers, thank you very much for your suggestion. We have made changes and improvements. "stress priming."That should be [30].

Round 2

Reviewer 2 Report

Comments and Suggestions for Authors

Well done. 

All the best

Reviewer 3 Report

Comments and Suggestions for Authors

Authors have addressed all the comments.